# Exosomes and Extracellular Vesicles in Myeloid Neoplasia: The Multiple and Complex Roles Played by These “*Magic Bullets*”

**DOI:** 10.3390/biology10020105

**Published:** 2021-02-02

**Authors:** Simona Bernardi, Mirko Farina

**Affiliations:** 1Department of Clinical and Experimental Sciences, University of Brescia, Bone Marrow Transplant Unit, ASST Spedali Civili, 25123 Brescia, Italy; mirkfar@gmail.com; 2Centro di Ricerca Emato-Oncologica AIL (CREA), ASST Spedali Civili, 25123 Brescia, Italy

**Keywords:** extracellular vesicles, exosomes, hematological malignancies, myeloid neoplasia, leukemia, AML, CML, MPNs

## Abstract

**Simple Summary:**

Extracellular vesicles (EVs) are released by the majority of cell types and can be isolated from both cell cultures and body fluids. They are involved in cell-to-cell communication and may shuttle different messages (RNA, DNA, and proteins). These messages are known to influence the microenvironment of cells and their behavior. In recent years, some evidence about the involvement of EVs and exosomes, an EV subgroup, in immunomodulation, the transfer of disease markers, and the treatment of myeloid malignancies have been reported. Little is known about these vesicles in this particular setting of hematologic neoplasia; here, we summarize and critically review the available results, aiming to encourage further investigations.

**Abstract:**

Extracellular vesicles (exosomes, in particular) are essential in multicellular organisms because they mediate cell-to-cell communication via the transfer of secreted molecules. They are able to shuttle different cargo, from nucleic acids to proteins. The role of exosomes has been widely investigated in solid tumors, which gave us surprising results about their potential involvement in pathogenesis and created an opening for liquid biopsies. Less is known about exosomes in oncohematology, particularly concerning the malignancies deriving from myeloid lineage. In this review, we aim to present an overview of immunomodulation and the microenvironment alteration mediated by exosomes released by malicious myeloid cells. Afterwards, we review the studies reporting the use of exosomes as disease biomarkers and their influence in response to treatment, together with the recent experiences that have focused on the use of exosomes as therapeutic tools. The further development of new technologies and the increased knowledge of biological (exosomes) and clinical (myeloid neoplasia) aspects are expected to change the future approaches to these malignancies.

## 1. Introduction

Extracellular vesicles (EVs) are natural membrane vesicles of 30–1000 nm in size. They can be isolated in vitro from cell-conditioned medium as well as from different body fluids (e.g., peripheral blood (PB), urine, breast milk) and have been recently recognized as being involved in intercellular communication. This mechanism is known to be essential in multicellular organisms and can also be mediated through direct cell–cell contact or transfer of secreted molecules [1,2]. EVs are loaded with different molecules, some of which are common to most EV subcategories released from the majority of cell types, such as the tetraspanin cluster of differentiation (CD; CD9, CD81, and CD63) and lactadherin [3]. Others, instead, are detected in EVs derived from only a specific subset of cell types. Their involvement in intercellular communication is due to their capability of carrying and delivering different markers from the ancestral cell, in particular, proteins, DNA, and RNA, to neighboring or distant cells [4]. For such reasons, EVs have been recently pointed out as one of the pivotal effectual players in liquid biopsy [5]. Moreover, EVs may even be engineered with synthetic therapeutic molecules, and, therefore, they have been presented as new therapeutic tools [4,6].

Based on their dimensions, EVs can be divided into different categories: microvesicles (MVs, 0.1–1 µm size) and exosomes (40–160 nm size), although larger EVs have been identified. Despite the fact that no consensus has yet been achieved, the International Society for EVs (ISEV) has recently grouped these particles as small EVs and large EVs; they have also reported additional criteria for EV classification (Table 1) [7]. Among EVs, exosomes are small EVs presenting a median diameter about ~100 nm and are endosomal- derived small vesicles [8].

Intercellular communication through exosome release seems to be involved in several physiological and pathological mechanisms [6,9], as mentioned above. In the last decade, exosomes have been investigated and associated with various disorders [10,11,12], including cancer [13,14], neurodegeneration [15], and inflammatory diseases [16,17]. The continuous evolvement of new technologies, such as next-generation sequencing (NGS) [18], digital PCR (dPCR) [19], and proteomic strategies [20], have enabled deep investigations of exosomes from different points of view. As a consequence, an increase in knowledge and discoveries concerning the real potential of these extracellular “magic bullets” has been observed. Their role in the pathogenesis and development of hematological malignancies has been recently demonstrated and progressively reported in hematologic neoplasia [21,22] originating from both lymphoid [23,24] and, later, myeloid lineage [25,26]. Myeloid neoplastic disorders are a group of very heterogeneous malignancies arising from the tumor transformation of a hematopoietic stem cell or a progenitor committed to myeloid differentiation [27]. Together with the great innovation of therapeutic strategies of myeloid neoplasia, the open study of circulating vesicles in different preclinical and clinical trials is expected to impact on the way we will evaluate minimal residual disease (MRD), identify new potential targets for therapy, and manage and deliver the treatments.

As a background to this review, we provide the main biological and clinical characteristics of myeloid malignancies in Table 2 and describe the results of some important studies on the impact on the changes induced to the microenvironment and on the immunomodulation supported by exosomes released by malignant myeloid cells. Afterwards, we present the potential use of exosomes as a carrier of biomarkers of myeloid neoplasia and their implication in the treatment of these malignancies, critically discussing the future perspectives of these approaches.

## 2. Small EVs and Their Impact on Microenvironment and Immunomodulation in Myeloid Neoplasia

Exosomes, as reported in the introduction, are endosomal-derived small vesicles released by most cell types; they are mainly involved in intercellular communication during physiological and pathological processes. Representing the “acellular copies” of the originating cells (including tumor and leukemia cells), exosomes contain a great variety of bioactive molecules; their ability to circulate in body fluids and their structure allow them to transport these molecules to distant targets. A large variety of studies showing that exosomes derived from malignant cells convey information not only between neoplastic cells but also to other cell types is now available [86]. The recipient cell types include different components of the immune system. There are different evidence suggesting that small vesicles released by tumor cells may contribute to malignant progression by influencing the immune system, likely skewing innate immune cells toward a favorable tumor progression phenotype [87]. Moreover, exosomes originated from neoplastic cells are considered a critical player in tumor and leukemia progression by creating a favorable microenvironment, delivering molecules modulating neovascularization and stroma [88,89]. Figure 1 represents a scheme of the potentiality of small EVs in myeloid malignancies. Crosstalk between leukemia cells resident in the BM and surrounding cells, including endothelial cells, is of pivotal importance for tumor growth in hematologic neoplasia, especially under hypoxic conditions. In fact, the hypoxia gradient is one of the major regulators of physiological and pathological hematopoiesis; it influences the content of EVs involved in angiogenesis in various types of hematological malignancies [90].

### 2.1. Chronic Myeloid Leukemia Small EVs on Microenvironment and Immunomodulation

Despite the great biological and clinical awareness about chronic myeloid leukemia (CML), little knowledge is available about the immunosuppressive and tumor-promoting properties of CML-derived exosomes. Recently, the K562 cell line served as an in-vitro model to better understand the real impact of exosomes. K562-derived exosomes are able to regulate the gene expression, cytokine secretion, and redox potential of BM mesenchymal stromal cells (BM-hMSCs) and murine macrophages. A recent study demonstrated that the expression of the genes involved in hematopoietic development and immune response regulation in BM-hMSCs were influenced by the time of exposure and exosome concentration. In particular, the authors reported an increase in TNF-alpha, interleukin-10, and nitric oxide levels in exosome-treated BM-hMSCs compared with the control. At the same time, reactive oxygen species production was downregulated in macrophages. Hence, CML-derived exosomes may alter the local BM microenvironment toward a leukemia-supporting niche by modulating inflammatory molecules and the redox potential of BM-hMSCs and macrophages [91]. Similarly, K562 CML-cell-derived exosomes have been reported to stimulate angiotube formation in a medium limited in serum and growth factors (*p* = 0.003) when internalized by human umbilical endothelial cells (HUVECs) during tubular differentiation, confirming their involvement in neovascularization. On the other hand, clinically active TKIs, such as imatinib and dasatinib, have been shown to reduce the total exosome release (*p* < 0.009) of K562, even if TKI-treated exosomes were, surprisingly, able to induce a similar tubular differentiation. However, HUVECs treated with dasatinib were markedly inhibited in their response to CML exosomes (*p* < 0.002). All the results were confirmed in vivo with BALB/c nude mice treated with subsequent injections of K562 exosomes and TKIs [92]. The same Italian research group investigated the exosome-mediated bidirectional crosstalk between leukemia and non-leukemia cells into a CML niche using another in-vitro model, the LAMA84 CML cell line, and newly diagnosed CML patients’ exosomes. The LAMA84-derived vesicles promoted the in-vitro adhesion of leukemic cells to a stromal monolayer and supported leukemic cell growth. Moreover, both the cell lines and the patients-derived exosomes activated epidermal growth factor receptor signaling in stromal cells, supporting the evidence of a microenvironment influenced by leukemia-derived small EVs, which promotes leukemic cell proliferation and survival [93]. Likewise, LAMA84 CML-cell-derived exosomes increased CML cell proliferation in a dose-dependent manner, as confirmed in a CML xenograft model. Therefore, exosomes are suspected of transporting molecules inhibiting apoptosis or inducing the expression of antiapoptotic molecules. TGF- β1 is considered the main character in these regulatory mechanisms because the exosome-mediated proliferation of leukemia cells and activation of antiapoptotic pathways can be inhibited by blocking TGF-β1 signaling [94]. Taken together, these studies strongly support the hypothesis of a microenvironment driven by CML-derived exosomes, affecting biological and clinical disease manifestation. In fact, it has been recently demonstrated that one way that leukemia cells resist TKIs is by taking advantage of signals from stromal cells in the BM. BM-hMSCs-derived exosomes contain fibroblast growth factor 2 (FGF2), a key player in this mechanism. FGF2-positive exosomes are subsequently endocytosed by myeloid leukemia cells, and this protects them from the effect of TKIs. Javidi-Sharifi, Martinez et al. observed that FGF2 and its receptor (FGFR) signaling were associated with increased exosome secretion and caused stromal growth. Thus, FGFR inhibition interrupts stromal autocrine growth and significantly decreases exosomal FGF2 production, resulting in less stromal protection of leukemia cells. Likewise, mice deficient in FGF2 and xenografted with *BCR-ABL1*-positive cells show increased overall survival when treated with TKIs [95].

This remarkable evidence highlights the great impact of FGF2-positive exosomes on CML leukemic niche immunomodulation and, in parallel, opens up new therapeutic options for overcoming TKI resistance by focusing on microenvironment control.

### 2.2. Acute Myeloid Leukemia Small EVs on Microenvironment and Immunomodulation

But what about other myeloid neoplasia? Are they simple manifestations of the aggressiveness of the malignant clone? Is the aggressiveness of the clone partially supported by a leukemia-devoted niche in myeloproliferative neoplasms (MPNs) and acute myeloid leukemias (AMLs) as well? Despite the use of interferon-alpha for therapeutic purposes in MPNs, to our knowledge, there is no evidence of the potential role of exosomes in immunomodulation or the induction of a leukemia-tolerating microenvironment in polycythemia vera (PV), essential thrombocythemia (ET), or myelofibrosis (MF). Conversely, some interesting data have been reported for AML-derived exosomes. The first investigation, based on the evidence that AML patients present decreased natural killer cytotoxicity, suggests an immunotolerance to leukemic cells. Microvesicles and exosomes from 19 newly diagnosed AML patients were analyzed in terms of numbers, molecular profile, and capability to suppress natural killer activity. AML patients’ PB contained an increased number of vesicles compared to healthy controls (*p* < 0.001), and, from what was observed in CML, TGF- β1 was confirmed as the key of this immunomodulatory mechanism. AML-patient-derived vesicles presented TGF- β1 and myeloid blasts markers, confirming their origin, and were able to decrease natural killer cell cytotoxicity in vitro (*p* < 0.002). Surprisingly, the coincubation of natural killer cells with microvesicles derived from AML patients neutralized anti-TGF-β1 antibodies and completely restored natural killer activity (*p* < 0.004) [96]. This result was fundamental because it proved the subversion of the AMLs microenvironment to support the leukemic persistence created by the exosomes; this was quickly confirmed by another research group. The authors found that exosomes isolated from both AML patients and four different AML cell lines contained various coding and noncoding RNAs that were relevant to leukemogenesis and AML progression. Furthermore, human leukemia cells, cultured across a 0.4-μm transwell membrane from murine BM-MSCs, were able to transfer leukemia-derived (human) transcripts to murine stromal cells via exosomes. Their uptake by BM-MSCs altered their proliferative, angiogenic, and migratory responses, proving the broad regulatory potential of the exosomes [97]. An elegant in-vivo study based on different mice models confirmed all these results. Kumar and colleagues preconditioned animal models with AML-derived exosomes and confirmed that they accelerated AML growth. This “priming” mechanism is due to the fact that AML-derived exosomes target BM stromal and endothelial fractions [98]. In BM-hMSCs, AML-derived exosomes induced the expression of normal hematopoiesis suppressors and a broad downregulation of hematopoietic-stem-cell-supporting factors [98]. This exosome-mediated downregulation of hematopoietic stem cells and progenitor cells may function directly (by targeted cell uptake) or indirectly (through niche stromal reprogramming) [26]. In particular, the trafficking of exosomal microRNAs (miRNAs) affects targeted cell activity and commitment. Exosomes isolated in vitro from HL60 and Molm-14 AML cell lines and in vivo from the plasma of AML-xenografted mice exhibited enrichment of miR150 and miR155. Exosomal miR150 and miR155 were able to impair the clonogenicity of hematopoietic stem cells, primarily by suppressing c-MYB translation. c-MYB is a transcription factor involved in healthy hematopoiesis; it regulates differentiation and proliferation. Moreover, miR155 appears to indirectly alter the actions of several proteins with established roles in malignancy, such as TP53 and BRCA1, and hematopoiesis, such as RUNX1 and MLL [99]. The same in-vitro/in-vivo model was investigated by Abdelhamed. AML-derived EVs, isolated in vitro from HL60 and Molm-14 cell lines, were able to induce protein synthesis inhibition and hematopoietic stem cell quiescence in murine BM. It was only partially revertible by BM transplantation. This effect was very similar to what was observed in AML patients when leukemia alters the hematopoiesis, causing anemia, fatigue, and uncontrolled infections. The authors demonstrated that EV miRNA cargo drove the mechanism by blocking different cellular functions and determining DNA damage [100].

Altogether, the reported studies have uncovered novel features of AML leukemogenesis and unveiled how AML cells induce a self-strengthening leukemic niche, promoting leukemic cell proliferation and survival while compromising normal hematopoiesis through exosome secretion. Finally, the studies have opened up new therapeutic approaches that focus on restoring natural immunological activity, for example, by targeting exosomal-TGF- β1, impairing natural killer cell cytotoxicity, or blocking miR155-positive exosomes.

An example of an experimental design to investigate exosome impairment of the immune system and the hematopoietic niche is graphically represented in Figure 2.

## 3. Small EVs as Myeloid Disease Markers

### 3.1. Small-EV Cargo as Disease Markers

Along with the investigations of their impact on the leukemic microenvironment and the immunological response, circulating exosomes’ cargo was deeply analyzed in order to identify potential leukemia markers [101]. Despite the recurring availability of leukemic cells in myeloid neoplasia in either the PB or in BM, several studies have been conducted in order to improve the sensitivity of the approaches and to reduce the number of invasive and painful procedures, such as BM aspirate or biopsy [33]. Thanks to the role played by the exosomes and following the promising results obtained in the solid tumors, these EVs have also been evaluated for their capability to be by themselves or to carry leukemic biomarkers. In principle, protein, lipid, miRNA, mRNA, or DNA profiles associated with different leukemias or malignancies can be identified in patients’ exosomes [102]. In the general scenario of myeloid neoplasia, the recent evidence of circulating exosomes as biomarkers of leukemia has introduced the potential for more sensitive noninvasive liquid biopsy approaches for the diagnosis, prognosis, and monitoring of leukemia patients [103]. In this context, the number of small EVs was reported to have a downward trend from diagnosis to the end of the therapy in responsive pediatric AML cases [104]. Similar evidence was reported in adult AML cases at diagnosis and different disease phases [96]. In this case, the authors reported a correlation between the leukemic burden and both the exosome number and their double-strand DNA (dsDNA) content [105]. Exosomal dsDNA has been reported as particularly informative when analyzed by NGS. Kontopoulou and colleagues identified a 90% overlap between the mutations identified in leukemic cell genomic DNA (gDNA) and exosomal dsDNA in pediatric AMLs. Moreover, they observed that the exosomal dsDNA mutations were no longer detectable after treatment in patients achieving complete remission [104]. A correlation between leukemic mutational status and exosome cargo in pediatric AML cases was also reported by the same research group. The vesicles’ transcript cargo was protected from degradation and was, therefore, particularly reliable. Exosomes’ RNA content was analyzed by GeneScan-based fragment-length analysis and real-time PCR assays, and the mutational status of *FLT3* and *NPM1* genes was compared with the gDNA. The small EVs’ content fully mirrored the leukemic mutational profile, supporting the potential use of exosomal RNA as a diagnostic biomarker in pediatric AML cases [106].

The exosomal transcript content was also investigated in myeloid malignancy, mostly analyzed and evaluated in terms of transcript value—the CML. Due to the paramount importance of MRD monitoring in this setting of patients, small EVs seemed to be the key to improving the sensitivity and reliability of resident leukemic cell detection. Some years ago, the scientific world of CML questioned the potential role of exosomes derived from CML cells as biomarkers and new targets for the detection of the *BCR-ABL1* transcript. The first approach was based on the use of ExoQuick™ Exosome Precipitation Solution for plasmatic vesicles isolation and nested PCR for *BCR-ABL1* transcript detection for CML patients. Exosomal RNA sequence analysis revealed 99% of homology with human cellular *BCR-ABL1*, even if only patients at the blast and accelerated phases showed the exosomal *BCR-ABL1* transcript [107]. Thanks to the advent of more sensitive and powerful technologies for both exosome isolation and transcript detection, such as dPCR [63,65], additional studies have been conducted. In particular, the feasibility of *BCR-ABL1* exosomal transcript detection in CML patients in the chronic phase, under treatment and with an undetectable level of MRD, has been reported [108]. This very important result has highlighted the possibility of exosome content analysis in order to improve the detection of active leukemic cells still resident in the patients’ BM.

The exosome transcript content in CML patients was further analyzed for miRNA cargo. The various roles played by miRNA in neoplasia have been continuously reported and confirmed [109]. Their potential use as circulating biomarkers has been widely speculated, either as cell-free miRNA or when included as vesicle cargo [110,111,112].

Among CML patients, a subset of them may benefit from TKI discontinuation, after which they may achieve “treatment-free remission” [56]. Musculoskeletal pain after stopping TKIs is a common manifestation. However, the possible factors related to this clinical condition are still matters for investigation. An analysis of circulating exosomal miRNA from CML patients that have undergone TKI discontinuation allowed the identification of a potential player. Exosomal miRNAs were profiled by TaqMan low-density array, which revealed exosomal miR140-3p to be significantly elevated in CML patients presenting musculoskeletal pain when compared to those without such pain (*p* = 0.0336) and to healthy subjects (*p* = 0.0022). miR140-3p is considered to have an inflammation-associated biological function, and its exosomal level is significantly decreased in CML patients that have experienced symptom relief. Thanks to these results, exosomal miRNA analysis may be a marker of therapy efficacy or side effects during TKIs treatment or suspension [113].

Exosomal miRNA content was also explored in AML patients with diagnostic, prognostic, and therapeutic monitoring purposes. miR155, a noncoding transcript of the B-cell integration cluster gene, has been described as a key player in the pathogenesis of several hematologic malignancies and to be part of exosomal cargo, as above reported [99]. For the first time, Caivano and colleagues investigated the potential clinical relevance of small EV miR155 levels. The authors observed that exosomal miR155 levels were correlated with a high white-blood-cell count in AML patients, and they were significantly higher in AML patients when compared to healthy controls (*p* = 0.01). Conversely, exosomal miR155 levels were significantly lower in MDSs. In addition, statistical analyses revealed significantly different small-EV miR155 patterns in AML cases compared to MDS cases (*p* = 0.04) [114]. Similarly, exosomal miR10b levels have been reported to have potential diagnostic impact on AML cases, independently from the cytogenetic risk. The expression levels of exosomal miR-10b, quantified by real-time quantitative PCR (RT-qPCR), were significantly higher in AML cases than in healthy controls and were strongly correlated with aggressive clinical characteristics. Moreover, statistical ROC curve analyses presented exosomal miR10b levels as a potential marker for the identification of AML patients from healthy controls (77.89% specificity and 82.50% sensitivity) [115]. Different from what was reported about miR155, exosomal miR10b expression levels were also related to prognostic significance. In particular, AML patients with higher exosomal miR10b expression levels presented significantly shorter survival, confirming exosomal miR10b as an independent prognostic marker for overall survival in AML cases [115]. Analogously, elevated exosomal miR125b levels were correlated with poor prognosis in adult intermediate-risk AML patients. In a cohort of 154 intermediate-risk AML cases, exosomal miR125b was prospectively quantified from diagnosis to the achievement of complete remission or to relapse. High exosomal miR125b levels were associated by different statistical tests with increased risks of relapse (*p* < 0.001) and overall death in 2 years (*p* < 0.001s) [116].

Another important exosomal miRNA, recently reported to be associated with myeloid malignancies, is miR532. This small noncoding RNA was reported to have prognostic value in many solid tumors [117,118]. Very recently, miR532 was quantified by RT-qPCR in exosomes isolated in 198 AML patients’ plasma. Despite what was previously reported for miR155, the authors did not observe any correlation between exosomal miR532 and hematological, clinical, and molecular characteristics. On the other hand, exosomal miR532 presented a prognostic value in both univariate and multivariate analyses. In particular, high exosomal miR532 levels were correlated with favorable overall survival [119], with an opposite trend when compared to exosomal miR10b [115]. Moreover, upregulation of miR-532 was negatively associated with a cellular metabolic profile, known to have a high impact on the clinical and biological characteristics of AMLs [120]. This biological evidence may explain and support the favorable prognostic significance of exosomal miR532 expression and its potential role as a survival predictor [119].

Overall, these important studies have highlighted the importance of the detection and quantification of circulating exosomal miRNA to improve the diagnosis and prognosis of myeloid neoplasia. Thanks to the recent development of innovative approaches, allowing the contemporary detection and analysis of multiple targets (e.g., NGS and RNA-seq) and the management of high-throughput data (e.g., artificial intelligence), the possibility of reliably valuing total exosomal miRNA content has become real. In this scenario, the contribution of Hornick and colleagues is particularly interesting. The authors reported the development of biostatistical models that are able to reveal circulating exosomal miRNA at low BM tumor burden, before peripheral blasts can be detected in AML patients, thus improving the early detection of AML recurrence [121].

As mentioned above, exosomes are known to have heterogeneous cargo, composed of nucleic acids (as described above) and also proteins and peptides. These molecules may act as disease biomarkers alongside DNA and RNA [16,122]. In myeloid malignancies, several studies have highlighted this aspect. Exosomal protein TGFβ-1 was one of the first to be investigated. It was highly expressed in adult AML patients’ exosomes, and TGFβ-1 exosomal expression was associated with a reduction in natural killer cell cytotoxicity (*p* < 0.002) [96], similar to what was reported for immunomodulation. Next, investigations demonstrated a correlation between exosomal TGFβ-1 protein levels and the response to induction (*p* < 0.004) and consolidation chemotherapy (*p* < 0.005) [123]. Therefore, exosomal protein content analysis may be eligible as a reliable tool for responses to therapy monitoring and may reflect the presence of residual leukemic cells in patients considered to be in complete remission. Might exosomal protein analysis also be informative for diagnostic purposes? A potential answer may be found in a recent trial investigating exosomal protein cargo in patients affected by PV. Despite the fact that the excess of EVs in these patients has been thought to contribute directly to thrombosis, the hypothetic role played by proteins contained in exosomes in thrombotic events in PV patients need to be confirmed. Using enzymatic-, mass spectrometer- and absolute quantification-based approaches, Fel and colleagues detected high concentrations of procoagulant and proangiogenic proteins in exosomes isolated from PV patients’ plasma [124]. These observations were immediately confirmed by an Italian study reporting small EVs released by megakaryocytes as a biomarker of thrombopoiesis in patients affected by MF (*p* < 0.001) [125]. In addition, the authors described, for the first time, a correlation between exosome numbers and myeloproliferative disorders, reporting the exosomes released by platelets to be increased in ET when compared to MF (*p* < 0.01) [125]. This last piece of evidence highlights the potential use of EVs and exosomes as disease markers as well as disease biomarker carriers, as stressed above. Further investigations may demonstrate a correlation between the exosomal procoagulant and proangiogenic protein contents and the risk of thrombotic events in myeloproliferative disorder patients. Moreover, this kind of analysis may support the development of more precise diagnostic procedures and target therapies. 

The analysis of markers shuttled in EV cargo, summarized in Table 3, has been reported as informative and important in order to improve the diagnosis and risk assessment of patients affected by myeloid neoplasia; However, it may not have been considered as an alternative to cell-based analysis. The results of EV analysis are expected to give additional information that is able to support clinicians through personal management and towards a better prognostication of patients.

### 3.2. Small EVs’ Membrane Proteins as Disease Markers

Although the pivotal and irrefutable importance of exosome cargo as a source of disease biomarkers in myeloid malignancies is clear, valuable significance might also be attached to molecular markers expressed on the exosomal surface. 

The heterogeneity of AMLs can be appreciated by the detection of cluster of CD combinations, both at diagnosis and during patient treatment. The membrane of exosomes derived from different AML cells is characterized by the presence of specific CD combinations, reflecting the cell of origin; the combinations may guide the molecular typing of AMLs in the clinic. Combining molecular biology techniques such as multiplex immuno-PCR and capillary electrophoresis with laser-induced fluorescence, the feasibility of simultaneous detection of a cluster of differentiation markers on AML-derived exosomes was very recently demonstrated [126]. The described approach presents a high detection sensitivity, up to picograms of protein concentrations, and adds a contribution to the possibility of monitoring exosomal membrane proteins as leukemic biomarkers. In addition, surface exosome proteins may also serve as targets in order to selectively isolated the microvesicles released by specific cell populations (Figure 3). 

One of the first malignant-derived exosome enrichment approaches was reported some years ago in AML adult patients. Taking advantage of an immunoaffinity-based capture method using magnetic beads conjugated with anti-CD34 antibodies, a CD34+ blast-derived exosome enrichment was tested. The levels of immuno-captured CD34+ exosomes correlated with the number of circulating CD34+ blasts in the PB of AML patients; additionally, CD34+ exosome cargo composition reflected the blast of origin molecular profile. This study provided a proof-of-concept of leukemia-derived exosome enrichment feasibility and suggested that a specific exosome immuno-capture might be useful in both the diagnosis and prognosis of AML patients [127]. Similar approaches were also explored in CML. In particular, Bernardi et al. investigated the feasibility of a leukemia-derived exosome enrichment using a commercial kit based on the immuno-capture of small EVs expressing a pan-cancer antigen. The reported approach received a head start from leukemia exosome enrichment and a *BCR-ABL1* transcript detection system based on dPCR [128]. For the first time, the authors demonstrated the detectability of *BCR-ABL1* transcripts in exosomes isolated from CML patients who were under treatment and presenting undetectable MRD levels [108,129]. Moreover, these *BCR-ABL1*-positive exosomes were shown to be useful in determining the molecular remission grade. Future studies are needed to demonstrate if they are better or worse than conventional approaches at predicting CML relapse [130]. 

In conclusion, we are now facing the real potential of small EVs in myeloid malignancies, and the advent of new technologies will enable us to constantly improve the sensitivity and precision of vesicular biomarker detection. However, standards need to be established for evaluating the composition and identity of small EVs, their isolation from body fluids, and the assessment of analytic methods. Moreover, further prospective clinical studies will better define the real impact of these shuttles on patient management.

## 4. Small EVs’ Involvement in the Therapy of Myeloid Neoplasia

### 4.1. Small-EV-Mediated Drug Resistance

Exosomes, and EVs in general, are known to be involved in drug resistance [131]. The first evidence of this arose from the study of circulating vesicles in solid tumors. Two of the principal mechanisms described are the delivery of chemo-resistant molecules [13,132,133] and the modulation of immunotherapy [134,135]. Successively, recent investigations have confirmed the presence of both mechanisms in myeloid malignancies as well, with special regard to AMLs. This discrepancy is not surprising, considering the quick onset of AMLs and their still-challenging treatment, due, in part, to the development of resistance and relapse. This fact is completely different from MPNs, with their chronic evolution, and CML, with its successful TKI-based treatment.

Chemotherapy resistance is established by clonal selection of resistant leukemia sub-clones. Careful analysis of leukemia-cell-derived exosome content has proved that vesicles are implicated in transferring several key mediators of chemo-resistance [114,136,137]. In particular, different elegant in-vitro models have analyzed the molecular pathways of exosome-mediated chemo-resistance in AML cell lines. At first, the promyelocytic leukemia HL60 cell line was studied using both the sensitive and multiresistant strains of the line. The latter strain overexpresses multidrug resistance protein 1. A chemo-resistance transfer from the multiresistant strain to the sensitive strain was observed when exposing sensitive HL60 cells to exosomes generated by resistant HL60. For the first time, this study revealed the capability of small EVs to horizontally transfer multidrug resistance, either by protein or nucleic acid delivery. In particular, the transferred exosomes were enriched in miR19b and miR20a, which have been suggested as circulating markers of chemo-resistance [138]. Nevertheless, related clinical trials have not yet been reported. Additional speculations may derive from two elegant subsequent studies correlating the modulation of the microenvironment with the exosomes that induce chemo-resistance. A coculture model of BM stromal cells and the KG1A AML cell line demonstrated that exosomes released by the different cocultured cell types protected AML cells against cellular death induced by etoposide, a drug molecule. In particular, the AML-cell-derived exosomes induce stromal cell IL8 production, which modulates the effect of etoposide. This impressive evidence describes the capability of AML cells to drive their microenvironment in response to treatment as well [137]. Similarly, AML-derived exosome-mediated angiogenesis has been related to vascular remodeling and chemo-resistance in vitro. Huang’s research team reported that AML-cell-derived exosomes, containing *VEGF* and *VEGF receptor* transcripts, bestowed neo-vasculogenesis and resistance to apoptosis enhancers in HUVECs [139]. 

Thus, these findings confirm the involvement of leukemia-derived exosomes in the acquisition of chemo-resistance; this may contribute to the introduction of new therapeutic strategies targeting exosomes in AML patients. In fact, the role of exosomes in treatment failure is still reported in clinical practice. A few years ago, Hong and colleagues described an exosome-mediated mechanism of interference with antileukemia functions of activated immune cells (NK-92) in a clinical trial. The authors demonstrated that exosomes released in patients’ plasma by AML cells blocked the antileukemia cytotoxicity of adoptive cell therapy and activated multiple suppressive pathways in the injected cells. AML exosomes reprogram NK-92 cells, reducing their therapeutic potential. Hence, plasma-derived exosomes collected in AML patients may also limit the expected therapeutic benefits of adoptive cell therapy [140]. Further studies may demonstrate the advantages of using antiangiogenic drugs, often ineffective in AML treatment, when targeting exosomes instead of leukemia cells.

Despite the aforesaid remarkable efficacy of TKIs in CML treatment, a small proportion of patients present drug resistance under TKI therapy. It is still a trigger for physicians. This phenomenon is mainly associated with acquired point mutations on the BCR-ABL1 protein, but little is known about the acquisition of resistance traits in cells not harboring these variants. One of the recently reported mechanisms is the exosome-mediated transfer of molecules from resistant to sensitive CML cells, comparable with what is observed in AML cells [138]. In particular, high levels of exosomal miR365 have been associated with lower drug sensitivity and a lower apoptosis rate. The exposure of sensitive CML cells to exosomes derived from resistant and miR365-rich induced drug resistance by inhibiting the expression of proapoptosis proteins in sensitive CML cells [141]. Conversely, miR328 has been reported as significantly associated with imatinib sensitivity in another in-vitro CML model. In particular, endogenous miR328 knockdown induced imatinib resistance in the K562 CML cell line, while in-vitro delivery of alkalized exosomes, with or without exosomal miR328, increased endogenous miR328 levels, sensitizing CML cells to imatinib [142]. Additionally, K562 imatinib sensitivity was tested in another study in the presence of exosomes released by human umbilical cord MSCs (hUC-MSCs). Exosomes derived from hUC-MSCs alone seem not to affect cell viability but promote imatinib-induced apoptosis and the activation of caspase-9 and caspase-3 when compared to imatinib alone. Hence, the synergy of exosomes and TKIs may be considered a promising approach to improve the efficacy of CML treatment [143].

Therefore, the summarized studies stress both general and specific drug-resistance that are mediated by exosomes and exosome cargo in myeloid neoplasia. Some of the in-vitro reported evidence needs to be confirmed by in-vivo models and in patients to estimate the real impact of circulating exosomes in drug-resistance manifestation. Nevertheless, it is irrefutable that there is a remarkable need to prove the utility of exosome analysis in refractory AML, MPN, and CML patients. This is of particular interest due to present and more ambitious treatment endpoints that require more sophisticated approaches to efficiently address drug resistance in a timely manner before clinical manifestation.

### 4.2. Small EVs as Potential Therapeutic Tools

The therapeutic effects of exosomes, and EVs in general, have already been extensively demonstrated, both when considering their natural properties and cargo and when engineered with drug molecules [144]. In particular, clinical trials focusing on exosome-based treatments have been reported in different clinical fields, such as cardiology [145,146,147], neurology [148,149,150], oncology [151,152,153], and immunology [154]. Along with all the other aspects mentioned in this review, the use of exosomes as therapeutic tools in oncohematology has only been recently explored and reported.

Interestingly, one of the first studies to test the immune-protective effect of exosomes secreted by an acute promyelocytic leukemia cell line (NB4) also studied exosome-based vaccines and antitumor immunotherapy reported for solid neoplasia. NB4-derived exosomes were used to pulse dendritic cells (DCs). Cytotoxicity tests demonstrated that cytotoxic T-lymphocytes activated by DCs stimulated with exosomes were more effective in neutralizing NB4 cells than cytotoxic T-lymphocytes activated by dendritic cells alone [155]. Exosome-based vaccines seemed to be very promising for successful AML management and treatment; these were further explored by other groups. In particular, the research team from the University of Campinas elegantly tested the feasibility of DC pulsing using exosomes isolated from AML patients’ plasma and AML cell line culture media. The authors confirmed the capability of exosomes in improving DC pulsing. Surprisingly, they observed that exosomes purified from culture media enhanced DC activation, inducing a higher level of target cell lysis than exosomes isolated from patients’ plasma [156]. Thus, exosomes from cultured cell media may represent an effective way to develop vaccines based on maturing DCs, even if recent data suggest that there is probably a synergic effect between vesicular and soluble molecules [157]. Considering the recent activation of anti-WT1 vaccine clinical trials for AMLs [158,159] and the results reported for antitumor vaccines [160], exosome-based vaccines are expected to be widely explored in the future, and the upcoming results would hopefully impact the way we manage AML cases.

But what about the use of exosomes as vehicles of drugs or therapeutic molecules? The use of exosome cargo for myeloid neoplasia treatment has been lukewarmly explored in AMLs and CML, while, to our knowledge, there are no data on MPNs. 

Among AMLs, one of the most interesting studies was based on the role of miR34c-5p in stem cells and senescence regulation. Low levels of miR34c-5p were described in leukemia stem cells and were associated with poor prognosis and response to AML therapy. The low levels of the investigated miRNA seem to be the effect of miR34c-5p exosome-mediated export. Despite this finely regulated escape mechanism, the forced increase in miR-34c-5p expression is able to induce senescence in leukemia stem cells. This is a very remarkable result, considering the key role of leukemia stem cells in relapse. The exosomal nature of miR-34c-5p suggests the potential use of small EVs to import miR-34c-5p in leukemia stem cells in order to promote senescence and establish new strategies for AML treatment by the tricky target of leukemia stem cells [161]. Later on, Dibavar and colleagues described an elegant study investigating the potential role of exosomes in promyelocytic leukemia. The authors exposed the NB4 cell line to BM-hMSC-derived exosomes isolated in-vitro by ultracentrifuge and arsenic trioxide, a conventional therapeutic molecule used in promyelocytic leukemia treatment. NB4 cells treated with exosomes and arsenic trioxide presented higher levels of apoptosis markers when compared with NB4 cells treated with exosomes or arsenic trioxide alone (*p* < 0.05) [162]. Hence, small vesicles should be new therapeutic tools to improve the efficacy and reduce the side effects of conventional and future drugs.

Notwithstanding the successful treatment based on TKIs, the investigation of exosome-based therapy was surprisingly more common in CML, combining the vesicles with both TKIs and unconventional molecules. An amazing study has described the target of CML blasts using engineered exosomes loaded with imatinib. The authors transfected the HEK293T cell line with plasmids encoding for exosomal protein Lamp2b, fused to a fragment of interleukin 3 (IL3) because the interleukin 3 receptor is known to be overexpressed on the surface of CML blasts. The exosomes released by the transfected cells were found to target CML cells and were able to deliver imatinib and *BCR-ABL1-*silencing RNA. This capability resulted in a reduction of leukemia cell growth in both in-vitro and in-vivo models and sensitive and resistant models. In particular, IL3-exosomes loaded with imatinib statistically reduced the tumor burden when compared with IL3-exosomes without imatinib, normal exosomes loaded with imatinib, and imatinib alone (*p* < 0.0005) in a mouse model [163]. These pivotal data strongly encourage the use of exosomes for specific drug delivery, even in CML, because of its efficacy in TKI-resistant cells as well. The authors obtained these impressive results after the previously reported experience of exosome delivery of unconventional molecules. In the CML context, the impact of curcumin on CML exosome composition [164] and the use of common lemon-juice-derived small vesicles have been reported. Lemon small vesicles were able to suppress CML tumor growth in xenograft models using NOD/SCID mice subcutaneously inoculated with CML cells. The exosomes specifically reached the tumor site and activated apoptotic cell processes [165]. 

Although very far from becoming clinical practice, the use of exosomes in the treatment of myeloid malignancies is no longer a theory. As potential markers of chemo-resistance, actionable drug-resistance mediators, or delivery vehicles of therapeutic molecules, exosomes have been demonstrated to possibly play a key role in the assessment of present and future myeloid neoplasia treatments. The future development of new drugs, together with additional knowledge about small vesicles and their manipulation, would strongly improve the impact of these extracellular shuttles and maybe result in a real revolution in the therapy of hematological neoplasia.

## 5. Conclusions

In this review, we presented and critically reviewed the most important knowledge about the impact of exosomes on myeloid neoplasia development, investigation, and treatment. Despite the majority of hematological studies being focused on exosomes in lymphoid neoplasia, as reported in a recent review [166], some evidence has unveiled the potential of those magic bubbles for malignancies derived from myeloid lineage as well. Both in order to improve the knowledge of these diseases and to open new delivery-dependent therapeutic strategies, exosomes may actually be novel allies for hematologists. Further large studies are needed to better understand how to transfer the results obtained in in-vitro or in-vivo models in clinical practice, especially for the use of vesicles as drug carriers. In parallel, the clinical application of leukemia-derived exosomes as disease markers seems closer. Some evidence of their detectability on plasma patients has already been described as encouraging results, even if future demonstration and standardization are undoubtedly needed. In particular, EV analysis may be complementary to leukemic cell investigations or conventional approaches, both at diagnosis and for MRD monitoring purposes. Based on the reported preliminary results. EVs should improve the prognostication and risk assessment and are expected to increase the sensitivity of resident active leukemia cells, as observed in CML, even if it is still a matter of debate. More advantages will probably be identified in the reduction of invasive biopsies such as BM sampling than in an immediate improvement of sensitivity and predictive value. In this context, the future development of new techniques and the wide distribution of novel biomolecular tools will surely support the discovery of new messages delivered by exosomes and the standardization of their application. We hope this review will contribute to extending the study of exosomes and EVs in the liquid counterpart of oncohematology and, in particular, myeloid malignancies.

## Figures and Tables

**Figure 1 biology-10-00105-f001:**
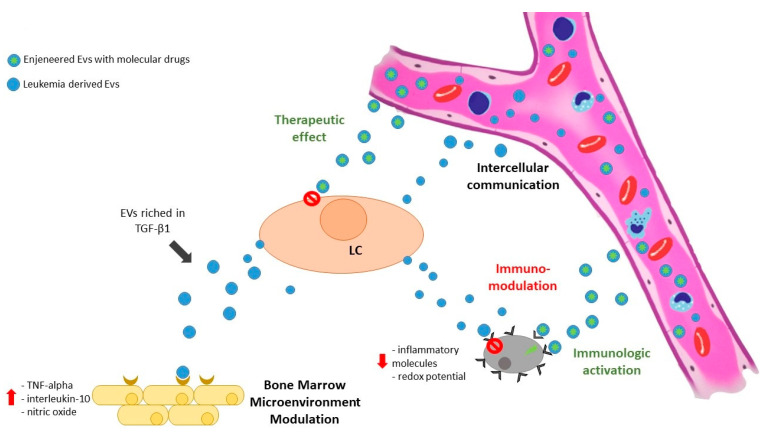
Exosomes released by leukemic cells may reach the microenvironment and modulate both stromal and immune cells. Moreover, leukemia-derived exosomes are able to target different cellular subpopulations, both close (by paracrine effect) and far (by circulating in biological fluids). On the other hand, exosomes may be used as therapeutic tools, shuttling drug molecules directly to leukemia cells or improving immune reactions by pulsing dendritic cells and T-cells. LC = leukemic cell; EVs = extracellular vesicles.

**Figure 2 biology-10-00105-f002:**
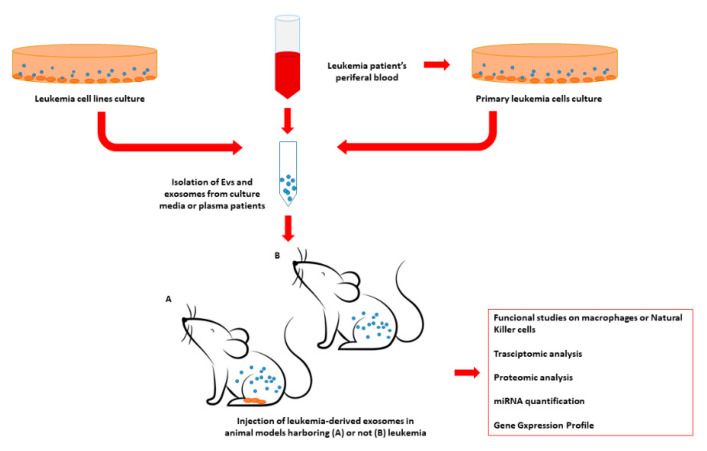
Example of experimental design for in-vivo assessment of the impact of leukemia-derived exosomes on the leukemic niche and the immune system. Exosomes may be isolated from commercial leukemia cell lines or primary leukemia cells derived from patient samples. In addition, exosomes may be isolated directly from leukemic patients’ plasma. Leukemia-derived exosomes may be injected into different animal models, such as murine models, presenting or not presenting leukemia. Subsequently, different tissue and cell populations isolated in the exosome-treated animals may be evaluated in terms of transcriptome, miRNome, protein expression, or serve as functional studies. EV = extracellular vesicles; miRNA = microRNA.

**Figure 3 biology-10-00105-f003:**
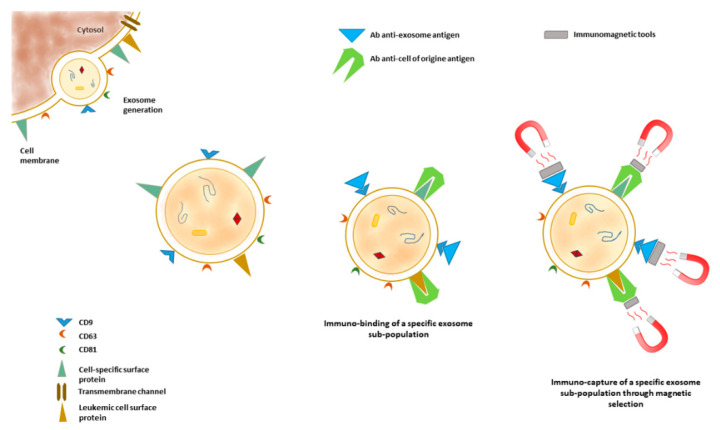
Biogenesis of exosomes and the presence of markers inherited from the cell of origin on the exosomes’ membrane. These markers can be targeted by peptides or antibody affinities. The use of peptides or antibodies conjugated with magnetic elements or engineered in order to be selectively sorted allows the selection of exosomes released by particular cell populations and increases the sensitivity and specificity of downstream analysis.

**Table 1 biology-10-00105-t001:** List of extracellular vesicles (EVs), classified following International Society for EVs (ISEV) 2018 guidelines [7].

Criteria	Classification
Size	Small EVs (<100 or <200 nm) Median/Large EVs (>200 nm)
Density	Low Medium High
Biochemical Composition	CD63+/CD81+ Annexin 5 Lactadherin
Origin	Endosome-origin “exosomes” Plasma-membrane-derived “ectosomes”
Condition or Cell of Origin	e.g., Podocyte EVs, Large oncosomes, Apoptotic bodies

**Table 2 biology-10-00105-t002:** Main biological and clinical characteristics of myeloid malignancies. AMLs = acute myeloid leukemias; MDSs = myelodisplastic syndromes; MPNs = myeloproliferative neoplasms; CML = chronic myeloid leukemia; PV = polycythemia vera; ET = essential thrombocythemia; MF = myelofibrosis; BM = bone marrow; PB = peripheral blood; BMT = bone marrow transplantation.

Disease	Main Characteristics	Main Phenotype	Clinical Presentation	Main Biological Features	Treatment	References
**AMLs**	Blocked or severely impaired differentiation of hematopoietic cells, resulting in a progressive accumulation of pathological cells (blasts) in various stages of incomplete maturation in bone marrow (BM)	Cytopenias in PB, sometimes leukocytosis	Asthenia and dyspnea (anemia), infections (neutropenia), and hemorrhages (thrombocytopenia)	Genetically heterogeneous. Molecular markers can be used to monitor MRD or for targeted therapy (e.g., FLT3, IDH1/2)	Chemotherapy, hypometilating agents, biological targeted drugs, BMT. Although the majority of patients have morphologic complete remission after they are treated with intensive chemotherapy, the relapse rate remains high.	[28,29,30,31,32,33,34,35,36,37,38,39,40,41,42,43,44,45]
**MDSs**	Clonal proliferation of hematopoietic stem cells, recurrent genetic abnormalities, myelodysplasia, ineffective hematopoiesis, PB cytopenia, and a high risk of evolution to AMLs	PB cytopenia (cytopenia in at least one line is a fundamental diagnostic criterion for MDS)	Asthenia (anemia), infections (neutropenia), and hemorrhages (thrombocytopenia)	Recurrent chromosomal abnormalities in about 50% of cases; cytogenetics + gene sequencing = 90% or more of patients. Useful for risk stratification and inform clinical decision-making.	The only curative treatment is the BMT. All patients with high-risk MDSs should be assessed for transplant eligibility. Most lower-risk patients do not need treatment immediately. Treatment options: hypomethylating agents to intensive chemotherapy or novel targeted therapies. The goal of treatment and support therapies is to ameliorate cytopenia and to improve the quality of life.	[46,47,48,49,50,51,52,53,54,55]
**Ph+ Myeloproliferative neoplasm**						
**CML**	The *BCR-ABL1* transcript deriving from t(9;22) deregulated tyrosine kinase is responsible for leukemic transformation and evolution, increasing proliferation, inhibiting apoptosis, and altering leukemic blasts’ adhesion to the BM niche	Leukocitosis (++ neutophils)	CML is characterized by a long first phase (chronic phase), followed by an increase in leukemic burden and progression (accelerating phase), resulting in a blast crisis, clinically resuming acute leukemia.	*BCR-ABL1* fusion gene resulting from t(9;22) reciprocal chromosomal translocation. *BCR-ABL1* transcript is the minimal residual disease marker. Mutations in BCR-ABL1 tyrosine kinase domains may drive therapy-resistance.	Three generations of tyrosine kinase inhibitors (TKIs) have been developed and are now available for treatment. Thanks to their remarkable efficacy, most CML patients present a normal life expectancy; an impressive proportion may even stop the treatment, achieving “treatment-free remission” (TFR)	[56,57,58,59,60,61,62,63,64,65]
**Ph- Myeloproliferative neoplasms (MPNs)**						
**PV**	Predominant erythroid proliferation	Erytrocitosis. Sometimes it could be associated with leukocytosys and thrombocytosis	Common signs and symptoms derived from microcirculatory disturbances (headache, itching, buzzing). Sometimes severe burning pain in the hands or feet that is accompanied by a reddish or bluish coloration of the skin. Higher risk of thrombosis.	*JAK2* V617F (95%); *JAK2* exon 12 mutations (5%). Subclonal mutations in myeloid genes can be found in patients with advanced disease and may lead to myelofibrotic or leukemic transformation	Phlebotomy and antiplatelet agents. Cytoreductive therapy (patients >60 years and/or personal history of thrombosis): hydroxyurea, interferon. JAK2 inhibitors for intolerant/unresponsive to hydroxyurea. Alkylating agents (second line)	[66,67,68,69,70,71,72,73,74,75,76,77]
**ET**	Overproduction of platelets (thrombocytes) by megakaryocytes in the BM	Thrombocytosis. Some patients present increased white blood cell count. A reduced red blood cell count has also been observed	The most common symptoms are bleeding, blood clots (e.g., deep vein thrombosis or pulmonary embolism), fatigue, headache, nausea, vomiting, abdominal pain, visual disturbances, dizziness, fainting	*JAK2* V617F (60–65%); *MPL* exon 10 mutations (5%); *CALR* exon 9 (20–25%); Triple-negative (5–10%). Subclonal mutations in myeloid genes can be found in patients with advanced disease and may lead to myelofibrotic or leukemic transformation	Low risk: Antiplatelet agents High risk: cytoreductive therapy: hydroxyurea, interferon, anagrelide.	[74,75,78,79,80]
**MF**	Clonal myeloproliferation, cytokine deregulation, BM reticulin and collagen fibrosis, risk of leukemic transformation	Splenomegaly (85%); cytopenias (2/3 of patients had anemia at diagnosis): some patients present with leukocytosis (40–50%) or thrombocytosis (13–31%).	Spleen-related symptoms: abdominal discomfort, early satiety, and pain under the left ribs. Constitutional symptoms (night sweats, fever, and cachexia). Asthenia (anemia) and hemorrhages (thrombocytopenia)	*JAK2* V617F (60–65%) *MPL* exon 10 mutations (5%) *CALR* exon 9 (20–25%) Triple-negative (5–10%). Subclonal mutations in *ASXL1, DMT3A, EZH2, IDH1/IDH2*, *SRSF2*, or *TP53* are associated with a worse clinical course and a higher risk of progression to blast phase or leukemic transformation	JAK1/2 inhibitors BMT Interferon (young, low-risk patients; during pregnancy). Cytoreductive therapy with hydroxyurea, alkylating agents. New biological targeted drugs: e.g., imetelstat. Treatment of anemia: erythroid-stimulating agents, steroids, androgens, or immunomodulatory drugs, which include thalidomide and lenalidomide.	[27,79,81,82,83,84,85]

**Table 3 biology-10-00105-t003:** Vesicular markers detectable in patients affected by myeloid malignancies and their potential application. AML = acute myeloid leukemia; CML = chronic myeloid leukemia; MPNs = Ph negative myeloproliferative neoplasms; MRD = minimal residual disease; EVs = extracellular vesicles.

Myeloid Neoplasia	Vesicular Markers	Purpose
**AMLs**	*FLT3* mutations	Diagnosis
	*NPM1* mutations	Diagnosis
	miR155	Diagnosis
	miR10b	Diagnosis and Prognosis
	miR125b	Prognosis
	miR532	Prognosis
	TGFβ-1 protein	Therapy response evaluation
**CML**	*BCR-ABL1* transcript	MRD monitoring
	miR140-3p	Therapy response evaluation
**MPNs**	Circulating EV number	Diagnosis and Risk stratification

## Data Availability

Not applicable.

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
