# Peer review of "Exosomes and Extracellular Vesicles in Myeloid Neoplasia: The Multiple and Complex Roles Played by These “Magic Bullets"

_biology, 2021, doi:10.3390/biology10020105_

Round 1

Reviewer 1 Report

The manuscript entitled “Exosomes and Extracellular Vesicles in the myeloid neoplasia: the multiple and complex roles played by these “magic bubbles” by Bernardi and Farina aims to review the functions of EVs in myeloid malignancies. The authors further discuss recent research into the possibility of exploiting vesicles for diagnostic and prognostic biomarkers in addition to therapeutic applications. While the article is conceptually interesting and covers a niche scientific field warranting discussion, several points should be considered prior to publication.

Major points:

  1. Section 2 includes a lengthy clinical introduction to myeloid neoplasms including MPNs, MDS, and AML. This section could be significantly shortened to focus on the relevant information pertaining to the pathophysiology regulated by EVs.
  2. Section 3 may be better organized into subsections detailing specific cell types in the leukemic microenvironment altered by EVs.
  3. Overall organization of the text could be improved by adding a summary sentence at the beginning or end of each paragraph.
  4. The figures are focused on broad concepts or laboratory techniques to study EVs. Additional detail including graphic representation of specific mechanisms of intercellular communication, biomarker candidates, or potential therapeutic applications discussed in the text would be valuable additions to the manuscript.
  5. The authors should refer to secreted vesicles as “extracellular vesicles” according to Minimal information for studies of extracellular vesicles 2018 (MISEV2018) standards (PMID: 30637094) unless the origin of vesicle biogenesis (e.g. bona fide endosomal-derived exosomes) is well characterized in the primary studies cited.
  6. A more balanced discussion of the limitations of EVs as useful biomarkers and therapeutic agents in hematopoietic neoplasms is warranted. For example, there is limited utility of EVs in the diagnosis of circulating neoplasms, as cell-based assays are certainly sensitive enough at the time of diagnosis when a large burden of circulating disease is present. The authors may consider focusing more on emphasizing the potential application of EV-based biomarkers for minimal residual disease detection. Even so, it is uncertain if increasing sensitivity of MRD using EV-based assays correlates with better prediction of prognosis or relapse – these limitations are briefly mentioned in line 543 and should be expanded upon.
  7. On a similar note, the EV isolation techniques used in the biomarker studies cited should be listed (e.g. in a table or supplemental data), as EV purification methods may have a large impact on the sensitivity or specificity of these assays.

Minor points:

  1. The terms “malicious” and “magic bubbles” may be reconsidered and omitted or re-phrased in this scientific review article.
  2. Both subsections of 4 (Exosomes as myeloid disease markers) are labeled 4.1.
  3. Discussion and citation of the following reference may be considered in section 3, on page 8: Abdelhamed et al. EMBO Rep. 2019. PMID: 31267709.
  4. Line 50: “Based on their dimension EVs can be divided into different categories.” The authors should clarify that currently EV populations are generally classified based on subcellular origin (e.g. plasma membrane-derived microvesicles, endosomal-derived exosomes) rather than size or dimension.

Author Response

The manuscript entitled “Exosomes and Extracellular Vesicles in the myeloid neoplasia: the multiple and complex roles played by these “magic bubbles” by Bernardi and Farina aims to review the functions of EVs in myeloid malignancies. The authors further discuss recent research into the possibility of exploiting vesicles for diagnostic and prognostic biomarkers in addition to therapeutic applications. While the article is conceptually interesting and covers a niche scientific field warranting discussion, several points should be considered prior to publication.

We really thank the Reviewer for all the comments and the deep revision of the manuscript. We detailed the modifications below. Please, note that the lines numbers refer to the tracked change version of the manuscript.

Major points:

Section 2 includes a lengthy clinical introduction to myeloid neoplasms including MPNs, MDS, and AML. This section could be significantly shortened to focus on the relevant information pertaining to the pathophysiology regulated by EVs.

Thank you for this important suggestion. According to Reviewers’ comment, the section has been organized in a summary table. Please see Table 2.

Section 3 may be better organized into subsections detailing specific cell types in the leukemic microenvironment altered by EVs.

Overall organization of the text could be improved by adding a summary sentence at the beginning or end of each paragraph.

Thank you so much. We considered the Reviewer’s suggestion. The involved cell types are macrophages, BM-hMSCs and natural killer cells. The majority of the studies investigated more than one cell type. For this reason, we divided the paragraph in two sub-sections based on myeloid neoplasms: chronic vs acute leukemias. Please, see the paragraph entitled “Small-EVs and their impact on microenvironment and immunomodulation in myeloid neoplasia”.

The figures are focused on broad concepts or laboratory techniques to study EVs. Additional detail including graphic representation of specific mechanisms of intercellular communication, biomarker candidates, or potential therapeutic applications discussed in the text would be valuable additions to the manuscript.

Thank you for this point. As suggested by Reviewers and Editor, the Figures were modified and we added specific markers and alterations, in particular in Figure 1. Please see Figure 1 and Figure 3.

The authors should refer to secreted vesicles as “extracellular vesicles” according to Minimal information for studies of extracellular vesicles 2018 (MISEV2018) standards (PMID: 30637094) unless the origin of vesicle biogenesis (e.g. bona fide endosomal-derived exosomes) is well characterized in the primary studies cited.

We agree with the Reviewers and revised the manuscript in order to unify the terminology used to indicate extracellular vesicles throughout the whole text and referred to MISEV2018. Please, see the modifications all over the text and the Introduction section Lines 50-54.

A more balanced discussion of the limitations of EVs as useful biomarkers and therapeutic agents in hematopoietic neoplasms is warranted. For example, there is limited utility of EVs in the diagnosis of circulating neoplasms, as cell-based assays are certainly sensitive enough at the time of diagnosis when a large burden of circulating disease is present. The authors may consider focusing more on emphasizing the potential application of EV-based biomarkers for minimal residual disease detection. Even so, it is uncertain if increasing sensitivity of MRD using EV-based assays correlates with better prediction of prognosis or relapse – these limitations are briefly mentioned in line 543 and should be expanded upon.

On a similar note, the EV isolation techniques used in the biomarker studies cited should be listed (e.g. in a table or supplemental data), as EV purification methods may have a large impact on the sensitivity or specificity of these assays.

Thank you for this pivotal suggestion. We tried to balance the conclusion of the review and of “Small-EVs cargo as disease markers” paragraph. Unfortunately, as reported in the text, only pan-cancer antigens or under-patent methods have been reported in EVs isolation in myeloid neoplasia. These are expected to improve the sensitivity and the minimal residual disease detection, but few data are available in order to assess a table. For this reason, we remark the concept in the text and stress this element in the discussion. Please, see lines 797-804.

Minor points:

The terms “malicious” and “magic bubbles” may be reconsidered and omitted or re-phrased in this scientific review article.

Thank you for this point. We substituted the term “malicious” with “leukemic” and magic bubbles as “magic bullets”, as suggested by Editor.

Both subsections of 4 (Exosomes as myeloid disease markers) are labeled 4.1.

Thank you and sorry for the mistake. The subsections are now correctly labeled 4.1 and 4.2.

Discussion and citation of the following reference may be considered in section 3, on page 8: Abdelhamed et al. EMBO Rep. 2019. PMID: 31267709.

Thank you for this suggestion. We added the reference in section 3 and discussed it together with another manuscript reporting the same in vitro-in vivo models (EVs isolated from HL60 and Molm-14 cell lines and injected in mice). Please, see lines 393-400

Line 50: “Based on their dimension EVs can be divided into different categories.” The authors should clarify that currently EV populations are generally classified based on subcellular origin (e.g. plasma membrane-derived microvesicles, endosomal-derived exosomes) rather than size or dimension.

Thank you for this point. Following Editor and Reviewers’ suggestions, we created a Table clarifying the EVs classification, since no consensus has yet been achieved, and added a comment in the Introduction. For the additional table (Table 1) we followed the MISEV 2018 nomenclature. Please see Table 1.

Reviewer 2 Report

The work is very interesting and generally well structured. The information provided on exosomes well underscore the importance of these vescicles and their potential role in treatment of cancer. 

However, the paragraphs of chapter 2 should be revised and streamlined. 

Author Response

The work is very interesting and generally well structured. The information provided on exosomes well underscore the importance of these vescicles and their potential role in treatment of cancer. 

However, the paragraphs of chapter 2 should be revised and streamlined. 

Thank you for your comment. The chapter 2 was substituted with a summary table. Please see Table 2.

Reviewer 3 Report

In this review, the authors offer a deepen and rich description of the main myeloid malignancies, and of the role performed by the extracellular vesicles, especially exosomes, released by tumor cells in the regulation of tumor microenvironment and the immune responses. The authors also describe the value of tumor-derived extracellular vesicles and their cargo in the diagnosis, prognosis, and clinical management of these malignancies, as well as their involvements in drug resistance mechanisms and novel potential use as therapeutic tool both naturally, and after in vitro engineering.

English

In my opinion, a moderate/extensive English revising is necessary.

Structure

The review is well-structured. However it is very long and full of data. To help the reading and fix the key points, it would benefit from a few added summary tables. For example, in the second paragraph (The myeloid malignancies) I suggest the adding of a table showing the different myeloid malignancies described, with columns reporting the main hallmarks, type of diagnosis, current available therapies, and the corresponding references. In the fourth paragraph (Exosomes as myeloid disease markers) I suggest the adding of a table reporting the main exosomal markers described, and the role performed, with the corresponding references.

Language and vocabulary

I strongly encourage the authors to unify the terminology used to indicate extracellular vesicles throughout the whole manuscript. In fact, a lot of different terms (exosomes, nano-vesicles, vesicles, micro-vesicles) are used alternatively. However, it’s necessary to be very careful when using these terms because no consensus has yet been achieved, and this could be confusing, and could be contested by experts in the field. To date, the best nomenclature has been described by the MISEV 2018 guidelines

I strongly suggest a more appropriate use of the punctuation throughout the whole text, in order to ameliorate and make more fluent the reading.

I also suggest to uniform the verbal tenses. In fact, the authors use both the simple past and the simple present tenses, often in the same phrase. For example line 135 and 136 “Most of patients became (past) dependent on... and, therefore, they receive (present) red-cell transfusion”; line 290, 291 and 292 “Javidi-Sharifi, Martinez et al. observed (past) that FGF2 and its receptor (FGFR) signaling is associated (present) with increased exosome secretion and causes (present) stromal growth. Thus, FGFR inhibition interrupts (present) stromal autocrine growth and significantly decreases (present) exosomal FGF2”; lines 296 and 297 “These remarkable evidences highlighted (past) the great impact of FGF2 positive exosomes on CML leukemic niche immunomodulation and, in parallel, open (present) to..”; line 327 “This result is (present) fundamental because it proved (past)..”, and so on

Finally, I suggest to use the plural to collectively indicate cells, exosomes, cell lines, microRNAs, etc, instead of talking of cell, exosome…, as singular entities.

In my opinion, throughout the whole manuscript, there is a misused of the Saxon genitive. I suggest the authors to edit it.

Figures

Figure 1 should be cited pertinently throughout the text. For instance Figure 1 could be cited in paragraph 3

Concept

I personally disagree with the term “magic bubbles” to refer to extracellular vesicles. I suggest the authors to revise this appellative to highlight the potentiality of EVs in the diagnosis, prognosis, and clinical management of myeloid neoplasia

Line 52: “excreted by many cells by endosome”. Erratum: exosomes are nano-vesicles released by multivesicular bodies, which in turn, derive from endosomes. Or they are nano-vesicles with an endosomal origin, or they are endosomal-derived nano-vesicles (as you properly wrote in line 239). Please correct

Paragraph 2.1: Please clarify if the acronyms given in the title (AML and MSD) are used as singular or plural. In fact, in this paragraph both AML and MSD are associated with verbs at both the third person singular (e.g., line 95 “AML leads”, line 102 “AML includes”) and the third person plural (e.g., line 89 “AML are characterized”, line 118 “MSD are”)

Moreover both AML and AMLs, and MSD and MSDs are used. Try to uniform the terminology. I suggest to use AML and MSD as singular, and AMLs and MSDs as plural.

Line 147: “Chronic myeloid leukemia (CML) is regarded as a paradigm of precision medicine” What means that CML (a disease) is a paradigm of precision medicine? Please clarify the concept

Lines 164 and 167: “BCL-ABL1 transcript” Please use the Italic font. In general, symbols for genes are italicized (e.g., IGF1), whereas symbols for proteins are not italicized (e.g., IGF1). The formatting of symbols for RNAs and complementary DNA (cDNAs) usually follows the same conventions as those for gene symbols.

Lines 180-182: “MPNs include Polycythemia Vera (PV), Essential Thrombocythemia (ET) and Myelofibrosis (MF). They are rare disorders, with increasing incidence with advancing age. The incidence in Europe is estimated at 1.8 cases/100,000 person-years” Please insert a citation

Line 233 and 234: “Representing the “acellular copies” of tumor and leukemia cells” Actually, this is true for all cells, including normal cells

Line 234: “impressive” In my opinion this adjective is inappropriate in this context

Line 236, 241, 276: “ Malignant exosomes” and “Malicious exosomes” What do you mean? Are they defined as malignant/malicious because derived from tumor/malignant cells, or are they malignant for themselves? Please rewrite the concept

Line 238: "Immune cell components" Do you mean components of an immune cell, or components (i.e. cells) of the immune system? Please correct

Line 246: “Hypoxia is a major regulator of exosomal content” What do you mean? How does hypoxia regulate the exosomal content? Please clarify the concept

Line 306: “different animal models” The figure shows a single animal model (mouse model). Please correct writing “such as…”

Lines 453, 454: “This small non coding RNA was reported with the prognostic value in many solid tumors” Please insert appropriate citations.

Lines 459: “high exosomal miR532 levels had a favorable overall survival” What do you mean? Maybe you wanted to write that “patients with high exosomal miR532 levels had a favorable overall survival”, or that “high exosomal miR532 levels were correlated with a favorable overall survival”. Please correct

Line 488: “Despite the excess of EVs in these patients has been thought to directly contribute to thrombosis” What do you mean? How excess of EVs contribute to thrombosis? Please clarify the concept

Typos

Lines 36, 325: “In vitro” Please use the Italic font

Line 37: “e.g. and such as” are synonymous. Please choose one of them

Line 52: “associated with” instead of “associated of”

Line 58: “From different points” instead of “Form different points”

Line 69: Targets instead of Target

Line 302: “in vivo”. Please use the Italic Font

Line 82: Biomarkers instead of Biomarker

Line 87: It would be more correct to talk of “Hematopoietic stem cells lineage”

Line 92: levels instead of level

Lines 96 and 200: “de novo” Please use the Italic font

Line 99: Becoming instead of became

Line 121: A instead of An

Line 125: “IPSS and IPSS-R” Please clarify these acronyms

Line 141: Go instead of goes (the subject, choices, is plural)

Lines 157 and 503: Targeted instead of Target

Lines 169 and 170: An instead of a, or sub-clone instead of sub-clones

Line 187: constitutive instead of constitute

Line 190: “the reminders constitutes” Subject and verb are not conjugated

Line 191: Mutations instead of mutation

Line 215: increasing instead of increase

Line 223: “only a small group of patients are eligible” Subject and verb are not conjugated

Line 259: The instead of that

Line 267: Were instead of was

Line 303: “May isolated” It would be more correct to write “may be isolated” Please correct

Line 353: Has instead of had

Line 419: “miRNA were” Singular or plural?,. Please clarify

Line 423: Exosomal level, instead of exosomal content

Line 424: “this results” Singular or plural? Please clarify

Line 432: Count instead of Counts

Line 434: “Significantly” It should be an adjective and not an adverb. Please correct

Line 440: As in line 423

Line 479: Exosomal instead of exosomes

Line 482: As in line 479

Line 533: suggests instead of suggest

Line 555: Mechanisms instead of mechanism

Line 580: Modulates instead of modulate

Line 640: DCs instead of DC

Line 641: As in line 640

Line 677: Successfull instead of succesfully

Author Response

In this review, the authors offer a deepen and rich description of the main myeloid malignancies, and of the role performed by the extracellular vesicles, especially exosomes, released by tumor cells in the regulation of tumor microenvironment and the immune responses. The authors also describe the value of tumor-derived extracellular vesicles and their cargo in the diagnosis, prognosis, and clinical management of these malignancies, as well as their involvements in drug resistance mechanisms and novel potential use as therapeutic tool both naturally, and after in vitro engineering.

We really thank the Reviewer for all the comments and the deep revision of the manuscript. We detailed the modifications below. Please, note that the lines numbers refer to the tracked change version of the manuscript.

English

In my opinion, a moderate/extensive English revising is necessary.

Structure

The review is well-structured. However it is very long and full of data. To help the reading and fix the key points, it would benefit from a few added summary tables. For example, in the second paragraph (The myeloid malignancies) I suggest the adding of a table showing the different myeloid malignancies described, with columns reporting the main hallmarks, type of diagnosis, current available therapies, and the corresponding references. In the fourth paragraph (Exosomes as myeloid disease markers) I suggest the adding of a table reporting the main exosomal markers described, and the role performed, with the corresponding references.

We agree with the Reviewer’s suggestion and substituted the paragraph 2 with a summary table about the main characteristics of the myeloid malignancies. Moreover, we added a Table concerning the main vesicular markers discussed in paragraph 4. Please, see Table 2 and Table 3.

Language and vocabulary

I strongly encourage the authors to unify the terminology used to indicate extracellular vesicles throughout the whole manuscript. In fact, a lot of different terms (exosomes, nano-vesicles, vesicles, micro-vesicles) are used alternatively. However, it’s necessary to be very careful when using these terms because no consensus has yet been achieved, and this could be confusing, and could be contested by experts in the field. To date, the best nomenclature has been described by the MISEV 2018 guidelines

Thank you for this point. According with Editor and Reviewers’ comments, we added a table concerning the possible classification of EVs and referred to MISEV 2018 in the introduction. Please, see line 50-54, Reference number 7 and Table 1. Unfortunately, we are not able to punctually define the type of EVs analyzed in the different cited studies because it was not well defined or investigated. For this reason, we limited the definition to the available information. E.g. small-vesicles.

I strongly suggest a more appropriate use of the punctuation throughout the whole text, in order to ameliorate and make more fluent the reading.

I also suggest to uniform the verbal tenses. In fact, the authors use both the simple past and the simple present tenses, often in the same phrase. For example line 135 and 136 “Most of patients became (past) dependent on... and, therefore, they receive (present) red-cell transfusion”; line 290, 291 and 292 “Javidi-Sharifi, Martinez et al. observed (past) that FGF2 and its receptor (FGFR) signaling is associated (present) with increased exosome secretion and causes (present) stromal growth. Thus, FGFR inhibition interrupts (present) stromal autocrine growth and significantly decreases (present) exosomal FGF2”; lines 296 and 297 “These remarkable evidences highlighted (past) the great impact of FGF2 positive exosomes on CML leukemic niche immunomodulation and, in parallel, open (present) to..”; line 327 “This result is (present) fundamental because it proved (past)..”, and so on

Finally, I suggest to use the plural to collectively indicate cells, exosomes, cell lines, microRNAs, etc, instead of talking of cell, exosome…, as singular entities.

In my opinion, throughout the whole manuscript, there is a misused of the Saxon genitive. I suggest the authors to edit it.

Thank you for all the above suggestions concerning the language. We tried to improve the consistency of the verbal tenses and of the plural forms. Moreover, a final English revision was performed. Please, see the tracked change version of the manuscript all over the text.

Figures

Figure 1 should be cited pertinently throughout the text. For instance Figure 1 could be cited in paragraph 3

We agree with the Reviewer and thank for this suggestion. We moved the Figure 1 in paragraph 3. Please see line 266.

Concept

I personally disagree with the term “magic bubbles” to refer to extracellular vesicles. I suggest the authors to revise this appellative to highlight the potentiality of EVs in the diagnosis, prognosis, and clinical management of myeloid neoplasia

Thank you for this point. We modified the term according with Editor and Reviewers’ suggestion.

Line 52: “excreted by many cells by endosome”. Erratum: exosomes are nano-vesicles released by multivesicular bodies, which in turn, derive from endosomes. Or they are nano-vesicles with an endosomal origin, or they are endosomal-derived nano-vesicles (as you properly wrote in line 239). Please correct

Thank you for this important point. We modified the text reporting the exosomes as endosomal-derived nano-vesicles. Please, see line 56

Paragraph 2.1: Please clarify if the acronyms given in the title (AML and MSD) are used as singular or plural. In fact, in this paragraph both AML and MSD are associated with verbs at both the third person singular (e.g., line 95 “AML leads”, line 102 “AML includes”) and the third person plural (e.g., line 89 “AML are characterized”, line 118 “MSD are”)

Thank you for these comments. Following Editor and Reviewers’ suggestion, we substitute this paragraph with a summary table. Please, see Table 2.

Moreover both AML and AMLs, and MSD and MSDs are used. Try to uniform the terminology. I suggest to use AML and MSD as singular, and AMLs and MSDs as plural.

We thank the Reviewer for this note. Since AMLs and MDSs are group of disease, we modified the terms as plural form in all cases, with the expect of results or evidences obtained from the study of a single type of AML or MDS.

Line 147: “Chronic myeloid leukemia (CML) is regarded as a paradigm of precision medicine” What means that CML (a disease) is a paradigm of precision medicine? Please clarify the concept

Lines 164 and 167: “BCL-ABL1 transcript” Please use the Italic font. In general, symbols for genes are italicized (e.g., IGF1), whereas symbols for proteins are not italicized (e.g., IGF1). The formatting of symbols for RNAs and complementary DNA (cDNAs) usually follows the same conventions as those for gene symbols.

Lines 180-182: “MPNs include Polycythemia Vera (PV), Essential Thrombocythemia (ET) and Myelofibrosis (MF). They are rare disorders, with increasing incidence with advancing age. The incidence in Europe is estimated at 1.8 cases/100,000 person-years” Please insert a citation

Thank you for these comments. Following Editor and Reviewers’ suggestion, we substitute this paragraph with a summary table. Please, see Table 2.

Line 233 and 234: “Representing the “acellular copies” of tumor and leukemia cells” Actually, this is true for all cells, including normal cells

We agree with Reviewer and specified that this characteristic is not limited to EVs released by malignant cells. Please, see line 255.

Line 234: “impressive” In my opinion this adjective is inappropriate in this context

We agree with Reviewer and modulated the sentence by omitting the adjective. Please, see line 256.

Line 236, 241, 276: “ Malignant exosomes” and “Malicious exosomes” What do you mean? Are they defined as malignant/malicious because derived from tumor/malignant cells, or are they malignant for themselves? Please rewrite the concept

We really thank the Reviewer for this point. We referred to EVs released by malignant cells in the different contexts. We re-wrote the sentences. Please, see lines 263-264.

Line 238: "Immune cell components" Do you mean components of an immune cell, or components (i.e. cells) of the immune system? Please correct

Thank you for this point. We modified the sentence as follow: “The recipient cell types include different components of the immune system”. Please, see line 260.

Line 246: “Hypoxia is a major regulator of exosomal content” What do you mean? How does hypoxia regulate the exosomal content? Please clarify the concept

We thank the Reviewer for this point. We referred to the importance of oxygen/hypoxia gradient in normal and pathological hematopoiesis and to the fact that the oxygen concentration influences the content of EVs involved in the cell-to-cell communication for angiogenesis activation/regulation. We re-phrased the sentence. Please, see lines 270-272.

Line 306: “different animal models” The figure shows a single animal model (mouse model). Please correct writing “such as…”

Thank you for this point. We correct the sentence accordingly writing “such as murine models…”, as suggested by Reviewer. Please, see line 415.

Lines 453, 454: “This small non coding RNA was reported with the prognostic value in many solid tumors” Please insert appropriate citations.

We apologized for the lack of citations. We added a reference. Please, see reference 119 and 120.

Lines 459: “high exosomal miR532 levels had a favorable overall survival” What do you mean? Maybe you wanted to write that “patients with high exosomal miR532 levels had a favorable overall survival”, or that “high exosomal miR532 levels were correlated with a favorable overall survival”. Please correct

We sorry for the evident error. We re-wrote the sentence as suggested. Please see “high exosomal miR532 levels were correlated with a favorable overall survival” at line 522.

Line 488: “Despite the excess of EVs in these patients has been thought to directly contribute to thrombosis” What do you mean? How excess of EVs contribute to thrombosis? Please clarify the concept

The excess of EVs is thought to contribute to thrombosis since it may improve the density of the peripheral blood. Moreover, EVs membrane may be characterized by the presence of proteins able to activate both the venous endothelium and the platelets. The concept is discussed in the cited reference number 126.

Typos

Lines 36, 325: “In vitro” Please use the Italic font

Line 37: “e.g. and such as” are synonymous. Please choose one of them

Line 52: “associated with” instead of “associated of”

Line 58: “From different points” instead of “Form different points”

Line 69: Targets instead of Target

Line 302: “in vivo”. Please use the Italic Font

Line 82: Biomarkers instead of Biomarker

Line 87: It would be more correct to talk of “Hematopoietic stem cells lineage”

Line 92: levels instead of level

Lines 96 and 200: “de novo” Please use the Italic font

Line 99: Becoming instead of became

Line 121: A instead of An

Line 125: “IPSS and IPSS-R” Please clarify these acronyms

Line 141: Go instead of goes (the subject, choices, is plural)

Lines 157 and 503: Targeted instead of Target

Lines 169 and 170: An instead of a, or sub-clone instead of sub-clones

Line 187: constitutive instead of constitute

Line 190: “the reminders constitutes” Subject and verb are not conjugated

Line 191: Mutations instead of mutation

Line 215: increasing instead of increase

Line 223: “only a small group of patients are eligible” Subject and verb are not conjugated

Line 259: The instead of that

Line 267: Were instead of was

Line 303: “May isolated” It would be more correct to write “may be isolated” Please correct

Line 353: Has instead of had

Line 419: “miRNA were” Singular or plural?,. Please clarify

Line 423: Exosomal level, instead of exosomal content

Line 424: “this results” Singular or plural? Please clarify

Line 432: Count instead of Counts

Line 434: “Significantly” It should be an adjective and not an adverb. Please correct

Line 440: As in line 423

Line 479: Exosomal instead of exosomes

Line 482: As in line 479

Line 533: suggests instead of suggest

Line 555: Mechanisms instead of mechanism

Line 580: Modulates instead of modulate

Line 640: DCs instead of DC

Line 641: As in line 640

Line 677: Successfull instead of succesfully

We modified the text according with all the above reported notes.

We really thank the Reviewer for the deep revision of our manuscript and all the comments, suggestions and notes. They really helped us in improving the quality of the review.

Round 2

Reviewer 1 Report

The majority of large concerns have been addressed by the authors. A few residual minor points:

1. Additional subsections may help break up long paragraphs of text and improve readibility, as previously recommended.

2. The authors should ensure that the term "magic bubbles" has been replaced throughout the manuscript (e.g. line 617).

3. The term "exosomes" is still used frequently, even though the context may indicate referral to extracellular vesicles in general. Please consider revising in accordance with MISEV2018 standards.

4. Moderate grammatical editing is still recommended.

Reviewer 3 Report

I have no suggestions.